# Risk Assessment of Avian Influenza Virus Subtype H7 Introduction and Spread in the Russian Federation

**DOI:** 10.3390/pathogens14111142

**Published:** 2025-11-11

**Authors:** Dmitry Varvashenko, Sergey Shcherbinin, Andrey Varkentin, Viktor Irza, Ilya Chvala, Alexander Sprygin, Mikhail Volkov

**Affiliations:** Federal Centre for Animal Health, 600901 Vladimir, Russia; sherbinin@arriah.ru (S.S.); varkentin@arriah.ru (A.V.); irza@arriah.ru (V.I.); chvala@arriah.ru (I.C.); sprygin@arriah.ru (A.S.); volkov_ms@arriah.ru (M.V.)

**Keywords:** avian influenza, H7 subtype, risk, probability of introduction, wild waterfowl, synanthropic birds

## Abstract

Avian influenza (AI) is a highly contagious viral disease affecting both domestic and wild birds, posing a significant threat to poultry farming worldwide. This study aims to analyze the key landscape and population factors associated with H7 avian influenza outbreaks across the Euro-Asian continent and to identify high-risk areas in Russia for the virus’s introduction and subsequent spread. Two models were developed using the Maximum Entropy algorithm (MaxEnt): An introduction model predicting the likelihood of avian influenza presence based on climatic, landscape, wild waterfowl and semiaquatic bird population density data; and a spread model estimating outbreak risk in poultry farms using data on synanthropic birds, poultry flock density, and proximity to wild bird habitats. The first model was trained via maximum likelihood using data from H7 avian influenza outbreaks in Europe (Italy, Germany, France, Denmark, Lithuania, the Netherlands) and Southeast Asia (China, Hong Kong, Taiwan, Japan, Cambodia, North Korea, South Korea, Vietnam). The second model was trained using output from the first model. Specifically, areas with a predicted probability of H7 outbreak between 0.9 and 1.0 were used as occurrence points for the model in Russia. The results demonstrated that both models achieved high predictive reliability for avian influenza outbreaks in the Russian Federation: the introduction model (AUC = 0.855) and the spread model (AUC = 0.993). Areas with a high probability of disease occurrence were identified in the Central, Southern, North Caucasian, and Volga Federal Districts. These findings underscore the necessity of enhanced disease surveillance in these regions, as well as in the border areas of the Ural, Siberian, and Far Eastern Federal Districts. The authors recommend strengthening biosecurity measures, enhancing wild bird monitoring in high-risk areas, and maintaining stocks of relevant vaccines to timely contain the outbreaks.

## 1. Introduction

The avian influenza virus (AIV), a type of influenza A virus, belongs to the genus *Alphainfluenzavirus* (*Influenzavirus A*), within the family *Orthomyxoviridae* and possesses a genome composed of eight segments of single-stranded negative-polarity RNA [1]. There are 18 known hemagglutinin (H) subtypes of the virus, with H5 and H7 subtypes being the most significant for birds [2,3]. The constant circulation of influenza viruses in both domestic and wild bird populations leads to genetic changes in the hemagglutinin (HA) protein, specifically at the cleavage site, which can alter the virus’s virulence. For H7 subtype influenza viruses, the presence of multiple basic amino acids at the hemagglutinin cleavage site is a key factor contributing to increased virulence [4,5,6,7]. In accordance with the recommendations of the Manual on Diagnostic Tests and Vaccines for Terrestrial Animals, outbreaks caused by avian influenza viruses of subtypes H5 and H7 are subject to notification to the World Organization for Animal Health (WOAH). While outbreaks of H7 avian influenza have not been reported in the Russian Federation, they remain a significant and ongoing concern in many neighboring countries, particularly the People’s Republic of China [8,9,10].

Avian influenza viruses, especially low-virulence variants, circulate widely in wild bird populations across the globe [11,12]. Moreover, several subtypes were identified in wild birds simultaneously [13]. Therefore, bird migrations play a crucial role in the spread of avian influenza (AI) viruses, acting as both a link in the natural disease cycle and a vector for introducing new low pathogenicity (LPAI) and high pathogenicity (HPAI) strains into previously unaffected areas. Birds can be asymptomatic carriers of avian influenza viruses, shedding the virus at their resting and nesting sites. This shedding contributes to genetic drift within the virus population, potentially leading to the emergence of new variants with altered biological characteristics. This can lead to large-scale reinfection of bird populations that were initially free from the virus, and can also trigger new outbreaks in wild animals [14,15,16]. Mechanical transfer of the avian influenza (AI) virus on feathers is a recognized means of transmission within avifauna, and is a subject of study in monitoring the disease [17]. But this may also be a possible introduction route via air inlets into poultry units.

The East Asia–Australasia Flyway Partnership (EAAFP) and the International Wader Study Group (IWSG) have identified nine global migratory bird flyways, with six passing through Eurasia, including Russia (Figure 1). Many birds undertake long-distance migrations, traveling from breeding grounds in the north to wintering grounds in the south and back again, with routes often connecting Russia to Europe, Africa, Australia, Asia, and North America [18,19,20]. There are reports indicating that low-virulent AIV variants can be spread over long distances by migratory birds, both within North America and between North America and Eurasia [21,22,23,24,25,26]. For our country, this is confirmed by the analysis of the bird ringing data in the Siberian and Northwestern Federal Districts, which suggests the migration of birds from all the above-mentioned continents through the Russian territory [27,28]. These data support the fact that various AIV subtypes can be introduced into Russia from all continents and subsequently spread to other countries.

The relevance of studying the introduction and transmission risks of avian influenza H7 subtype derives from its zoonotic potential—the virus demonstrates pathogenicity across avian, mammalian, and human hosts. Evidence comes from the 2013–2017 Chinese outbreak, where 1568 human cases were documented, 615 of which proved fatal [29].

To date, ecological niche modeling using the maximum entropy (MaxEnt) algorithm is a well-established technique for studying wildlife habitat dynamics [30,31,32]. Despite the MaxEnt benefits for ecological niche modeling, the maximum entropy method has been adapted and is also used in a number of publications to predict new outbreaks [33,34,35,36,37].

This paper presents a model for the potential introduction and spread of the exotic AIV H7 in Russia, incorporating landscape, population, and geographical risk factors.

## 2. Materials and Methods

### 2.1. Study Design

Two types of models were developed. The first model, which used landscape factors as predictors, also incorporated the population density of wild waterfowl and semiaquatic birds (hereafter classified as migratory birds) within specific biotopes. The model envisages suitable conditions for the disease outbreak occurrence (hereinafter referred to as the “introduction model”). The second model incorporated factors related to the presence of humans, domestic, migratory, and synanthropic birds, and envisaged the likelihood of AI outbreaks among poultry (hereinafter referred to as the “spread model”).

### 2.2. Data Sources

Data on AI H7 outbreaks, providing geographical coordinates (longitude, latitude), were obtained from the Empres-i database of the Food and Agriculture Organization of the United Nations (FAO) (https://empres-i.apps.fao.org (as of 21 May 2025)) [38]. The analysis includes only outbreaks notified by the WOAH. The study focused on avian influenza virus (AIV) H7 cases within wild bird populations and domestic ducks, specifically highlighting the role of ducks (*Anatidae*) as major players in the spread of the virus. Thus, the coordinates of 161 AI H7 cases were imported.

In order to study the spread of AIV H7 using migration patterns of wild birds, GenBank data was analyzed. The dataset was filtered by virus type (Influenza A virus), genomic region (nucleotide sequence of the H7 hemagglutinin gene), and publication date (sequences published between 1902 and 2025). All H7 sequences available in the GenBank database after filtering were analyzed (*n* = 3844). Subsequently, isolates recovered from poultry (*n* = 1830) were excluded from the analysis. The data obtained (*n* = 2014) allowed us to identify the wild bird species involved in the pandemic.

Next, the coordinates of individual bird species’ locations, documented as occurrence records, were imported from the GBIF (Global Biodiversity Information Facility) database [39,40,41]. The data was filtered according to three criteria:(1)A temporal range from 2005 to 2025, selected because this 20-year period encompasses reported peaks of H7-subtype avian influenza epidemics [8];(2)Key bird species known to be involved in avian influenza spread;(3)Records of bird locations sourced from naturalists, ornithologists, and camera traps.

We obtained more than 800,000 coordinates of *Anseriformes* location and more than 900,000 coordinates of *Charadriiformes* location [39,41]. After filtering out records with missing values, approximately 1.5 million bird sighting coordinates from both taxonomic orders were retained for analysis. Data for these orders were obtained for the entire study area.

For the second model, occurrence coordinates of synanthropic birds from the *Passeriformes* and *Columbiformes* orders within the Russian Federation were obtained from the GBIF database as key predictor variables. In total, more than 1,000,000 locations were obtained [40,42]. Following data processing, approximately 700,000 geolocation records for both avian orders were retained for analysis.

### 2.3. Hazard Identification

The study area covered the Russian Federation and Eurasian countries where H7 avian influenza has been reported over the past 20 years (Figure 2). Given the vast territory and diverse climatic conditions of the Russian Federation, this study incorporated climatic and ecological data from a range of countries in Europe (e.g., Denmark, France, Germany, the Netherlands, Italy, Lithuania) and Asia (e.g., China, Japan, both Koreas, Taiwan, Vietnam, Cambodia) for a comprehensive analysis. Given the absence of reported H7 avian influenza outbreaks in Russia, this study aimed to model both the probability of the virus’s introduction into the Russian Federation and the subsequent risk of its spread among domestic poultry populations.

### 2.4. Ecological Niche Modeling

AIV H7 ecological niche modeling and the subsequent assessment of the risk of its spread were performed using the maximum entropy algorithm (MaxEnt) [31]. A suite of predictive factors was selected based on their influence on virus dispersal by migratory birds, encompassing climatic, landscape, and population variables (Table 1). The selected variables have demonstrated their significance in similar scientific publications [33,34,35,36,37]. All predictor raster layers for the first model were pre-processed to a common coordinate system and a uniform spatial resolution (1 × 1 km). For the second model, the raster resolution was standardized to 10 × 10 km to match the spatial grain of the domestic chicken density data layer. The data on migratory and synanthropic birds were transformed into raster layers with the appropriate name and content. To achieve this, a ‘collector activity’ model was implemented. This algorithm, which calculates the saturation of unique events, is highly effective for generating density layers compatible with MaxEnt (version 3.4.4) software [32].

Alongside climatic and environmental variables, the distribution of migratory birds was incorporated into the introduction model. These factors indicate that the avian influenza virus H7could be introduced to Russia via wild waterfowl and semiaquatic bird migrations. The “spread model” was constructed using only population factors, an approach that implicates migratory and synanthropic birds in the potential spread of the influenza virus to poultry farms.

### 2.5. Model Construction and Validation

Model construction and validation were performed using an initial dataset of AIV H7 presence data and the predictor variables derived from Table 1. For the first model, we generated two sets of background points: 2000 localized points within a 5000 km radius of each presence point, and 1000 random background points distributed across the entire study area. The model construction produced output probabilities ranging from 0 to 1. The probability values were classified using the Jenks natural breaks optimization method. The classified probability values visualized the predicted spatial variation in risk. For the second model, presence points were defined as those assigned a probability between 0.9 and 1.0 by the first model. For the distribution model, 1000 background points were generated, distributed throughout Russia. The regularization multiplier β was set to 0.5 to generate less smoothed risk zones [32]. The predictive accuracy of the model was evaluated using the Area Under the Curve (AUC) metric. AUC values greater than 0.8 indicated good performance, values between 0.7 and 0.8 indicated general performance, and values below 0.7 indicated poor performance [48].

We assessed multicollinearity among the selected predictors using the Variance Inflation Factor (VIF). This factor allows for the identification of a high correlation between variables, which can compromise the stability and interpretability of regression models. Predictor values were extracted from tif raster files at each sample location to create the analysis dataset. We constructed linear models with a dummy response variable solely to calculate the VIF and assess multicollinearity, not to evaluate predictors’ impact on the actual outcome. The obtained VIF values were analyzed against standard thresholds (5 and 10) to identify potentially redundant variables and inform the decision to retain or exclude them from the model. This step ensures a more reliable construction of subsequent models, reduces the risk of overfitting, and enhances their generalization performance [49].

To assess variable importance, we performed a jackknife test. This involved alternately excluding each variable from the model and measuring the associated decrease in predictive performance.

### 2.6. Software

Geospatial processing, including mapping, was conducted in QGIS 3.42 [50]. Species distribution modeling was performed using a maximum entropy approach implemented in MaxEnt version 3.4.4. [30]. The entire analytical workflow, including model execution and result processing, was managed within RStudio (2025.05.0+496).

## 3. Results

The analysis of the hemagglutinin (HA) gene of AIV H7 included sequences isolated from 84 species of wild birds (Figure 3A). The bird species were classified into 13 orders. The major orders were *Anseriformes* (number of involved species *n* = 42) and *Charadriiformes* (number of involved species *n* = 12). AIV H7 was also detected in 12 synanthropic passerine species, with the Eurasian tree sparrow (*Passer montanus*) being the primary vector, accounting for six of the recovered isolates (*n* = 6). Notably, AIV H7 cases were also detected in pigeons (*Columbiformes*) (*n* = 15) (Figure 3B).

VIF analysis revealed no values exceeding the thresholds of 5 and 10, indicating an absence of multicollinearity among the variables (Figure 4). This confirms the model’s factor selection, ensuring that the subsequent conclusions are not distorted by multicollinearity.

Based on environmental and climatic data, our model visualized areas within Russia at high risk for the introduction of AIV H7 (Figure 5).

As shown in Figure 5, regions at potential high risk for AIV H7 introduction (R = 0.7–1.0) are located in the Central, Southern, North Caucasus, Volga Federal Districts, the Northern Arctic, the Left Bank of the Dnieper River, the Leningrad and Kaliningrad Oblasts, the Khanty-Mansi Autonomous Okrug and the border regions of the Siberian Federal District. The model also exhibited a high performance value, AUC (introduction) = 0.855. The most important variables for the introduction model were human footprint index and the type of land cover. The second most important variables were the altitude level, the density of migrating birds, the distance to water bodies, the average annual temperature and average annual precipitation (Figure 6).

We identified 256 points with a high risk of AIV H7 introduction (R = 0.9–1.0) from the first model’s output (Figure 7). The spread model exhibited excellent performance, AUC (spread) = 0.993. The most important variables for the spread model were: human footprint index and density of migratory and synanthropic birds (Figure 8).

The second model visualized the highest-risk areas for AIV H7 spread within Russia (Figure 9).

As shown in Figure 9, regions at a potential high risk of AIV H7 spread (R = 0.7–1.0) are located in the Northwestern, Central, Southern, North Caucasian, and Volga Federal Districts, the Left Bank of the Dnieper River, the Khanty-Mansiysk Autonomous Okrug, and the border regions of the Siberian and Far Eastern Federal Districts, taking into account the highest risk values in the regional capitals (R = 0.9–1.0).

## 4. Discussion

When analyzing the distribution of species involved in the AIV H7-caused epidemic process, representatives of the *Anseriformes* and *Charadriiformes* orders were predominant, which is consistent with the results of many years of research by scientists [12,13,14,51]. Among synanthropic birds, *Passeriformes* and *Columbiformes* were the dominant reservoirs in the spread of the avian influenza virus subtype H7. This is also consistent with other published data [15,16,52,53].

The study identified high-risk areas in the Russian Federation for the introduction of the H7 avian influenza virus by wild waterfowl and aquatic birds. Russia’s extensive water areas serve as vital stopover sites, refueling points, and wintering grounds for migrating birds. These are the Caspian Sea; the Black Sea; Lake Baikal; the basins of the Volga, Ob, Lena, Amur rivers. Birds use these areas not only as final destinations but also as stopover sites, as their migration flows are oriented toward nesting in more northern territories [15]. The results of the study by Zhang J.L. et al. [54], 2022, demonstrated that the accumulation of waterfowl in wintering areas correlates with AIV H7 spread, with migration being a key factor. Air sampled from wetlands with high concentrations of key bird species contained more H7 influenza virus particles than other areas [54].

The waning influence of variables related to water resource distribution may stem from their correlation with other critical factors, such as migrating bird density and proximity to water bodies. This may also be due to Russia’s more extensive hydrographic network compared to the other countries included in the study. Although these variables were not highly informative on their own, the model still detected high-risk areas in close proximity to lacustrine and riverine environments across Russia (Figure 10).

Poultry farms located near wild bird habitats and flight paths, the potential for direct contact between wild and domestic birds, and lax biosecurity measures significantly increase the risk of avian influenza virus subtype H7 spreading to these farms [55,56].

The relative contribution of the poultry density variable diminished, likely due to covariance with other, more influential predictors in the model. It can be concluded that the spread is impossible without a human factor (human activity). For example, the widespread existence of backyard and small-scale farms in Russia contributes to the constant presence of synanthropic birds there. The combined influence of other variables outweighs, but does not eliminate, the effect of poultry density. Consequently, viral spread remains likely following an introduction event, even in areas with low poultry density.

Thus, modeling results indicate a high risk for the introduction and spread of AIV H7 in several regions, including the Central, Southern, North Caucasus, and Volga Federal Districts, the Leningrad Region, the Left Bank of the Dnieper River, and border areas of the Ural, Siberian, and Far Eastern Federal Districts. This correlation exists because these areas have a high Human Footprint Index (0.7–0.9), indicating high densities of both wild and synanthropic birds near poultry operations.

Uncontrolled avian influenza outbreaks, especially in poultry farms, can lead to the release of a large amount of viral material into the environment, creating persistent viral reservoirs that facilitate the continued spread of the pathogen and pose risks to other farms and wildlife. Therefore, implementing strict virological surveillance of both migratory and domestic birds—across backyard, small-scale, and commercial farms—is essential, and must be coupled with enhanced biosecurity measures. An urgent priority is ensuring the availability of relevant vaccine stocks for the timely containment of disease outbreaks.

## Figures and Tables

**Figure 1 pathogens-14-01142-f001:**
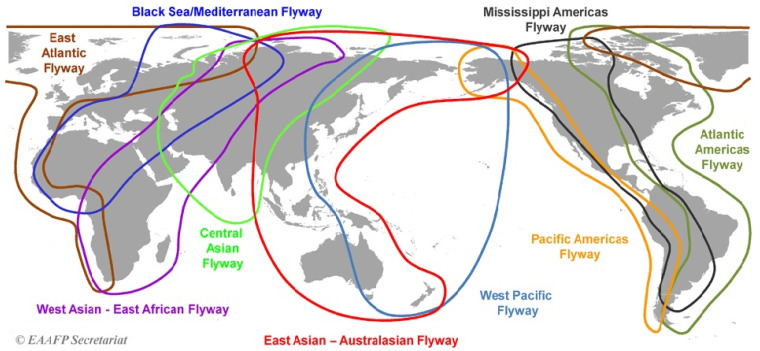
Largest flyways of migratory birds, according to the EAAFP Secretariat [20].

**Figure 2 pathogens-14-01142-f002:**
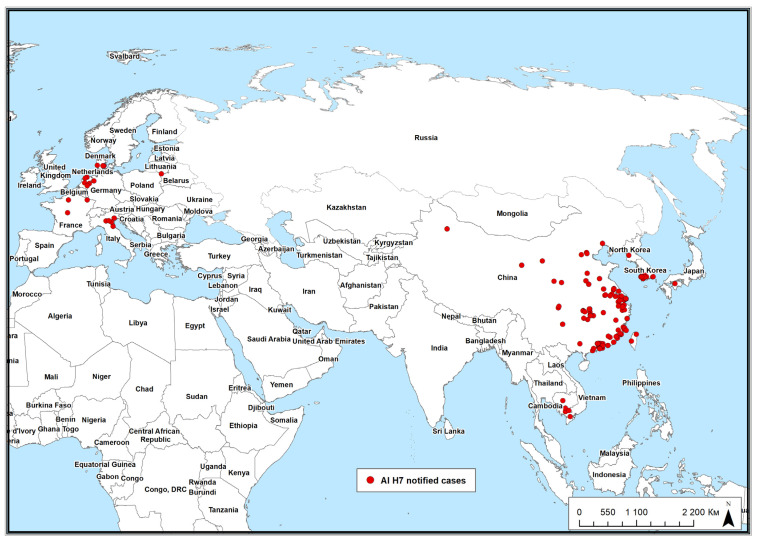
AI H7 notified cases in wild birds in 2005–2025 (according to Empres-i).

**Figure 3 pathogens-14-01142-f003:**
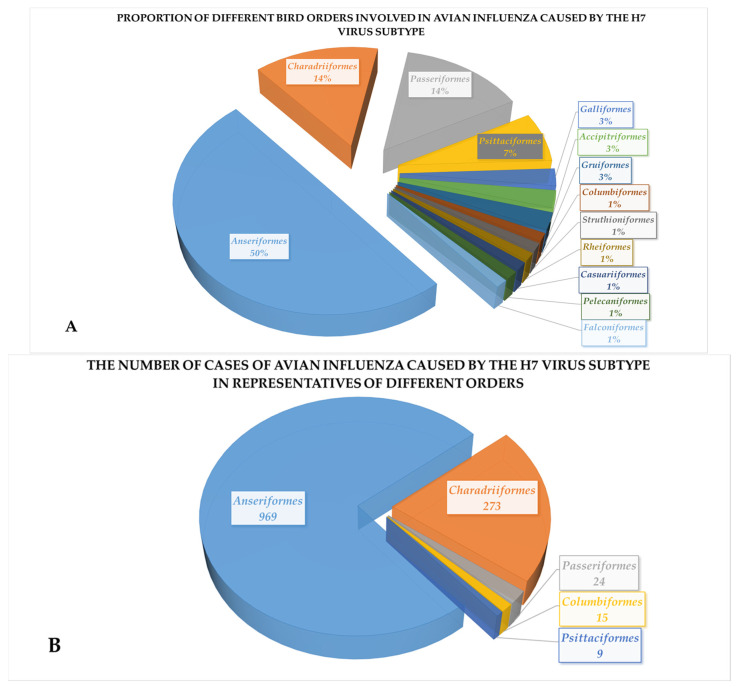
Biodiversity of wild birds involved in AIV H7 pandemics in 1902–2025: (**A**) Proportion of different bird orders involved in avian influenza caused by the H7 virus subtype; (**B**) The number of cases of avian unfluenza caused by the H7 subtype in representatives of different orders.

**Figure 4 pathogens-14-01142-f004:**
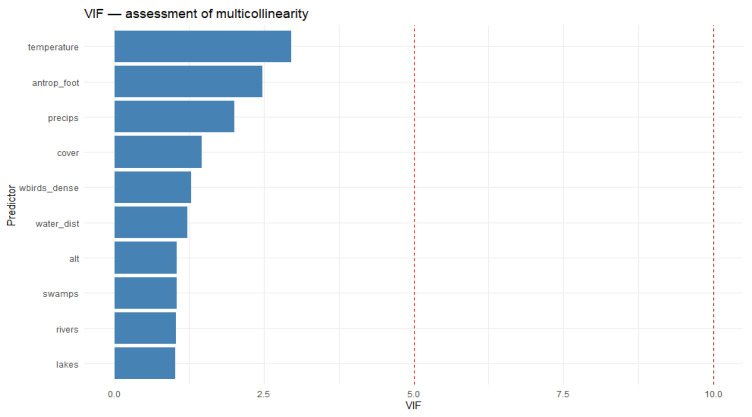
VIF-assessment of multicollinearity for the model variables.

**Figure 5 pathogens-14-01142-f005:**
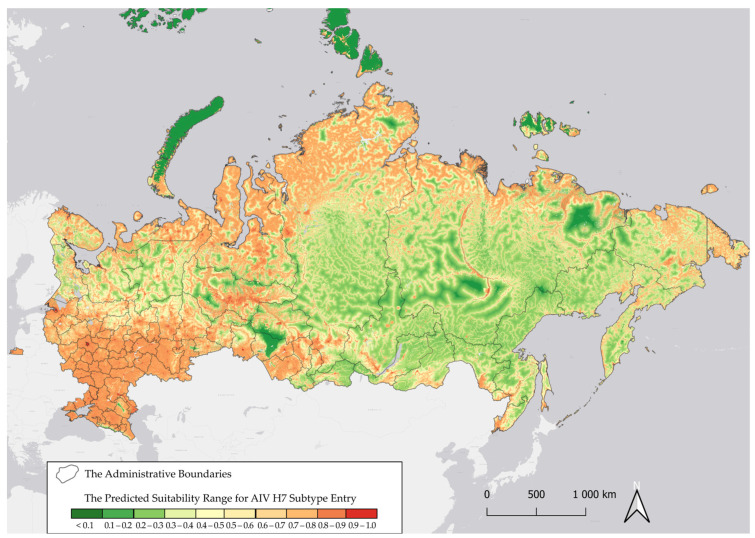
Predicted regions of Russia within at potential risk for the introduction of H7 avian influenza virus via wild migratory birds.

**Figure 6 pathogens-14-01142-f006:**
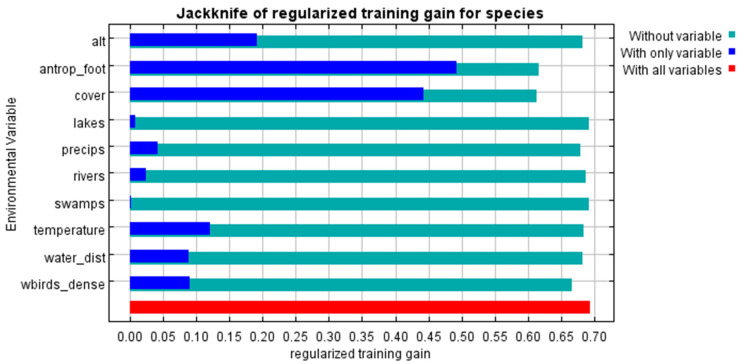
Jackknife Test of Variable Contributions for Modeling AIV H7 Introduction into Russia.

**Figure 7 pathogens-14-01142-f007:**
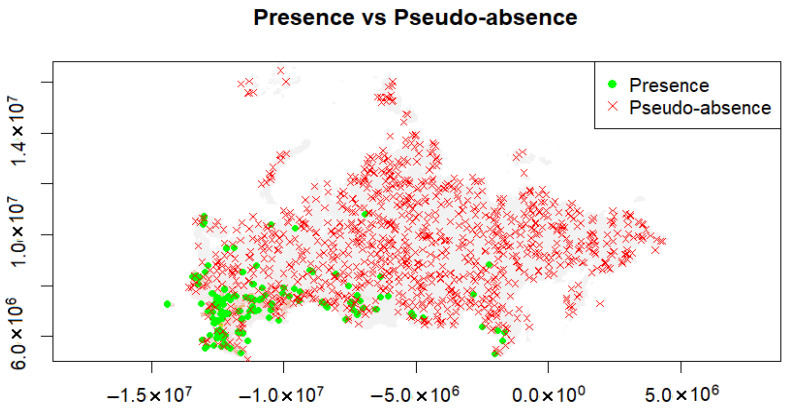
The presence points obtained from the introduction model and the distribution of background points used for the second model.

**Figure 8 pathogens-14-01142-f008:**
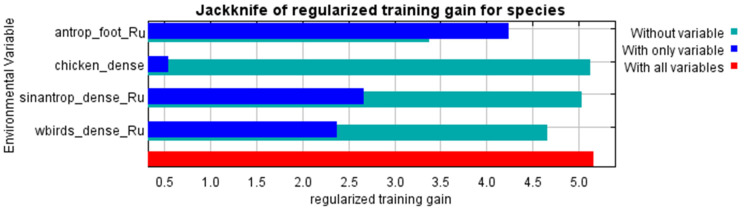
Jackknife Test of Variable Contributions for Modeling AIV H7 Spread in Russia.

**Figure 9 pathogens-14-01142-f009:**
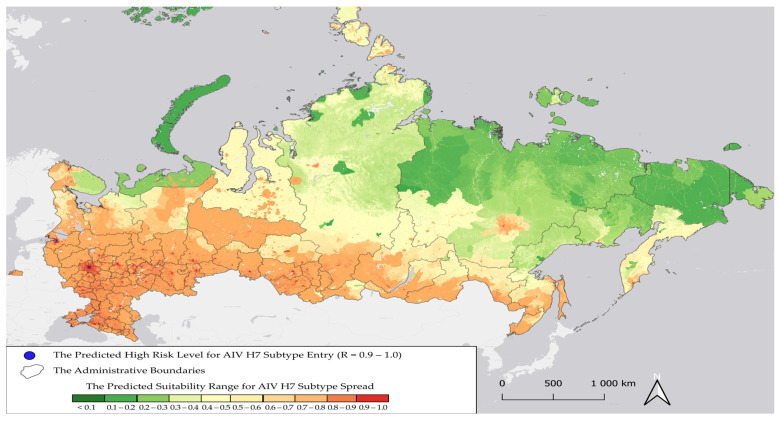
Predicted regions within the Russian Federation at potential risk for H7 avian influenza virus spread.

**Figure 10 pathogens-14-01142-f010:**
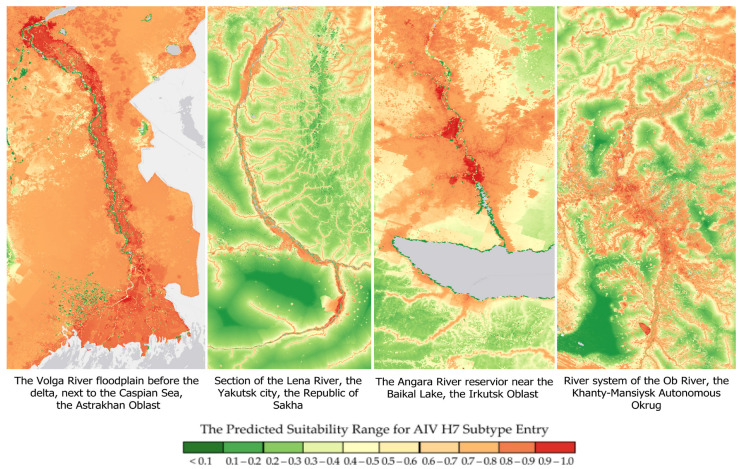
Areas near Russia’s largest water bodies predicted to be at high risk for avian influenza (AI) H7 outbreaks.

**Table 1 pathogens-14-01142-t001:** Predictor variables characterizing ecological and geographical conditions used for ecological niche modeling.

Variable	Meaning	Factor	Unit of Measurement	Reference
temperature	average annual air temperature	climatic	°C	https://www.worldclim.org/ (accessed on 25 May 2025) [43]
precips	average annual precipitation	climatic	millimeters	https://www.worldclim.org/ (accessed on 25 May 2025) [43]
lakes	distribution or presence of lakes	environmental	%	https://www.hydrosheds.org/ (accessed on 21 May 2025) [44]
swamps	presence and distribution of swamps	environmental	%	https://www.hydrosheds.org/ (accessed on 21 May 2025) [44]
rivers	river network or proximity to rivers	environmental	kilometers	https://www.hydrosheds.org/ (accessed on 21 May 2025) [44]
water_dist	distance to the nearest water bodies	environmental	kilometers	modeled in QGIS
alt	digital relief model (height above mean sea level)	environmental	meters	https://www.worldclim.org/ (accessed on 25 May 2025) [43]
cover	types of land cover	environmental	category	https://due.esrin.esa.int/ (accessed on 25 May 2025) [45]
wbirds_dense	population distribution or density of migrating birds in the study area	population	birds/km^2^	https://www.gbif.org/ru/ (accessed on 3 June 2025) [39,41]
wbirds_dense_Ru	population distribution or density of migrating birds in Russia	population	birds/km^2^	modeled in QGIS
sinantrop_dense_Ru	population distribution or density of synanthropic birds in Russia	population	birds/km^2^	https://www.gbif.org/ru/ (accessed on 1 June 2025) [40,42]
chicken_dense	population density of chickens	population	birds/km^2^	https://www.fao.org/ (accessed on 25 May 2025) [46]
antrop_foot	human footprint (impact of humans on the global environment)	Population	Index	https://doi.org/10.1111/gcb.14549 (accessed on 3 June 2025) [47]
antrop_foot_Ru	human footprint (impact of humans on the global environment) for Russia	Population	Index	modeled in QGIS

## Data Availability

The datasets generated during and/or analyzed during the currentstudy are available from the corresponding authors (Varvashenko D., Shcherbinin S.) upon reasonable request.

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
