# Peer review of "Risk Assessment of Avian Influenza Virus Subtype H7 Introduction and Spread in the Russian Federation"

_pathogens, 2025, doi:10.3390/pathogens14111142_

Round 1
Reviewer 1 Report
Comments and Suggestions for Authors
This study applies ecological niche modeling to assess the potential introduction and spread risk of H7 subtype avian influenza viruses (AIVs) across the Russian Federation and neighboring Eurasian regions. The authors integrated outbreak records, environmental and climatic variables, and migratory bird distribution data to construct two MaxEnt-based models (“introduction” and “spread”), providing a spatial framework for understanding the drivers of AIV transmission. The findings contribute to a better understanding of how wild birds and anthropogenic factors may shape the spatial dynamics of H7 AIVs. However, several issues concerning data presentation and contextual interpretation should be addressed as follows:
- The use of abbreviations should be standardized throughout the manuscript. In line 78, “MaxEnt” should be introduced as “Maximum Entropy (MaxEnt) algorithm” when first mentioned. In line 195, the correct abbreviation for “hemagglutinin” is “” Please ensure that all abbreviations are defined at first use and used consistently thereafter.
- The citation format in line 82 is inconsistent with the reference list. Please review and ensure that all in-text citations and reference entries follow a uniform style throughout the manuscript.
- In Figure 3, subtitles should be moved to the caption rather than embedded in the figure. The labels “A” and “B” should be positioned in the upper-right corners of each subfigure. In addition, all Latin names (e.g., Anseriformes, Charadriiformes, Passeriformes, Columbiformes) should be italicized.
- The maps in Figures 5 and 9 should include a north arrow and latitude–longitude gridlines to improve geographic precision and reproducibility.
- In Figure 9, the blue color used to represent “High Risk Level for Entry” lacks sufficient contrast with the background. Consider adjusting the color scheme to enhance clarity and readability.
- It is recommended to expand the Introduction with more background information specifically related to the H7 subtype avian influenza virus—such as its epidemic history, host range, and public health relevance—to make the introduction of the modeling study more coherent.
- Inrecent years, several novel avian influenza viruses emerged in the countries that covered by the East Asia–Australasia Flyway, indicating the increasing risk of wild bird originated viruses. In addition to the H7 viruses, the authors are encouraged to concern the critical role of migratory birds in reservation and dissemination of avian influenza viruses, including the highly pathogenic H5 viruses and the low pathogenic viruses, in the the Introduction. Because the current circulated H5N1 viruses were originated from their H5N8 ancestors, which were closely related with the transcontinental movements of migratory birds (Cui et al., PMID: 34757542). The rare subtypes were not frequently detected in domestic birds, but majorly harbored by migratory birds, such as Anseriformes and Charadriiformes (Shen et al., PMID: 39248597).
- The English language is generally accurate; however, some sentences are overly long and could be simplified to improve fluency and readability.
Author Response
Comments 1: The use of abbreviations should be standardized throughout the manuscript. In line 78, “MaxEnt” should be introduced as “Maximum Entropy (MaxEnt) algorithm” when first mentioned. In line 195, the correct abbreviation for “hemagglutinin” is “HA” Please ensure that all abbreviations are defined at first use and used consistently thereafter.
Response 1: We appreciate your comment. We concur with your observation and have incorporated it into the revised version.
Comments 2: The citation format in line 82 is inconsistent with the reference list. Please review and ensure that all in-text citations and reference entries follow a uniform style throughout the manuscript.
Response 2: Thank you for the comment. A typographical error resulted in an unintended space between the sources, which we have now rectified.
Comments 3: In Figure 3, subtitles should be moved to the caption rather than embedded in the figure. The labels “A” and “B” should be positioned in the upper-right corners of each subfigure. In addition, all Latin names (e.g., Anseriformes, Charadriiformes, Passeriformes, Columbiformes) should be italicized.
Response 3: We appreciate your observation. This has been duly considered in the editing process.
Comments 4: The maps in Figures 5 and 9 should include a north arrow and latitude–longitude gridlines to improve geographic precision and reproducibility.
Response 4: Thank you. We have added a north indicator to the maps. In consideration of the latitude-longitude grid, we have ascertained that the incorporation of these elements would impair the visual clarity and readability of the figure.
Comments 5: In Figure 9, the blue color used to represent “High Risk Level for Entry” lacks sufficient contrast with the background. Consider adjusting the color scheme to enhance clarity and readability.
Response 5: We are grateful for your observation. We recognize that the blue colouration in the figure may present challenges for visual interpretation. However, in our high-resolution rendering, the blue exhibits sufficient contrast with the red component. For optimal viewing, we can furnish the figures in their original high-resolution version.
Comments 6-7: It is recommended to expand the Introduction with more background information specifically related to the H7 subtype avian influenza virus—such as its epidemic history, host range, and public health relevance—to make the introduction of the modeling study more coherent...
Response 6: We are grateful for your valuable recommendation. As a result, we have broadened the introductory section of the manuscript.
Reviewer 2 Report
Comments and Suggestions for Authors
Interesting paper.
Line 61-62: Mechanical transfer of the avian influenza (AI) virus on feathers is a recognized means of transmission within avifauna and is a subject of study in monitoring the disease [24]. But this may also be a possible introduction route via air-inlets into poultry units.
Line 75: … into Russia from all continents, and subsequent spread to other countries.
Line 99-106: … and domestic ducks….. Please explain and argument why you would limit this to only domestic ducks?? There are other poultry species (layers, broilers, turkeys…..) that have been infected by HPAI H7 subtype. I see now plausible reason why you would limit the study to only domestic ducks.
Line 109-121: For the first model you limit the wild bird species to Anseriformes and Charadriformes? What is the rationale behind this??
Line 122-126: and the same for the second model: why limit the wild bird species to Passeriformes and Columbiformes? What is the rationale behind this??
Line 147-148: …the domestic chicken density data layer. You mention here a broader range of poultry (chickens) then what you mention in line 99 ?? And it aligns with my comments about line 99, that in my opinion it is strange that you limit the study only to domestic ducks as poultry species.
Line 213 (text accompanying the Figure). This a very sparse explanation of what this Figure means. I would recommend to replace it by: Figure 4. VIF-assessment of multicollinearity for the model variables.
Line 217: With respect to figure 5: the text accompanying figure 5 mentions predicted regions within the Russian Federation …… But in your map you do also show prediction of areas in Europe, China, Japan…… So you have to change the text under Figure 5; and may be change the figure in a way comparable to Figure 9. Furthermore it would be sensible to mark the boarder of the Russian Federation in the map with a thick line.
Line 262: …study by Zhang et al. [21], demonstrated that ……..
Author Response
Comments 1: Line 61-62: Mechanical transfer of the avian influenza (AI) virus on feathers is a recognized means of transmission within avifauna and is a subject of study in monitoring the disease [24]. But this may also be a possible introduction route via air-inlets into poultry units.
Response 1: We are grateful for your valuable comment. As per your suggestion, the recommended sentence has been integrated into the manuscript.
Comments 2: Line 75: … into Russia from all continents, and subsequent spread to other countries.
Response 2: We appreciate your feedback. The proposed sentence has been duly added to the text.
Comments 3: Line 99-106: … and domestic ducks…..Please explain and argument why you would limit this to only domestic ducks?? There are other poultry species (layers, broilers, turkeys…..) that have been infected by HPAI H7 subtype. I see now plausible reason why you would limit the study to only domestic ducks.
Response 3: Thank you for raising this question. Notably, in Russia and East Asian regions, domestic ducks (Anatidae family) are commonly reared in free-range systems. This husbandry method renders them a pivotal node in the epidemiological transmission chain, owing to their increased potential for interaction with wild and synanthropic bird populations.
Comments 4: Line 109-121: For the first model you limit the wild bird species to Anseriformes and Charadriformes? What is the rationale behind this??
Response 4: Thank you for raising this point. Literature evidence, further substantiated by our own research data (referenced in lines 200–206), identifies Anseriformes and Charadriformes as the key bird orders implicated in the transmission dynamics of the pathogen.
Comments 5: Line 122-126: and the same for the second model: why limit the wild bird species to Passeriformes and Columbiformes? What is the rationale behind this??
Response 5: Very interesting question! Literature and observational data confirm that members of the Passeriformes and Columbiformes orders function as key synanthropic reservoirs and transmitters of avian influenza virus in the context of internal (within-country) disease spread.
Comments 7: Line 213 (text accompanying the Figure). This a very sparse explanation of what this Figure means. I would recommend to replace it by: Figure 4. VIF-assessment of multicollinearity for the model variables.
Response 7: We appreciate your editorial input. In accordance with your suggestions, we have revised the title of Figure 4.
Comments 8: Line 217: With respect to figure 5: the text accompanying figure 5 mentions predicted regions within the Russian Federation …… But in your map you do also show prediction of areas in Europe, China, Japan…… So you have to change the text under Figure 5; and may be change the figure in a way comparable to Figure 9. Furthermore it would be sensible to mark the boarder of the Russian Federation in the map with a thick line.
Response 8: Thank you for the remark. In light of this, we have updated the title of Figure 5.
Comments 9: Line 262: …study by Zhang et al. [21], demonstrated that ……..
Response 9: Thanks for the note! Fixed the error.
Round 2
Reviewer 2 Report
Comments and Suggestions for Authors
I have still two problems with your response:
Comments 3: Line 99-106: … and domestic ducks…..Please explain and argument why you would limit this to only domestic ducks?? There are other poultry species (layers, broilers, turkeys…..) that have been infected by HPAI H7 subtype. I see now plausible reason why you would limit the study to only domestic ducks.
Response 3: Thank you for raising this question. Notably, in Russia and East Asian regions, domestic ducks (Anatidae family) are commonly reared in free-range systems. This husbandry method renders them a pivotal node in the epidemiological transmission chain, owing to their increased potential for interaction with wild and synanthropic bird populations.
Reaction to response: it is true that outdoor-raised poultry has an increased potential for interaction with wild bird populations. But chickens in East Asian regions are also raised in open sheds with potential contact with wild birds. Furthermore, in West-European countries, poultry farms with chickens and domestic ducks get also infected although there is no physical contact with wild birds because the poultry is housed in closed poultry units. So in my view there is no reasonable argument to limit the analysis to only domestic ducks.
Comments 5: Line 122-126: and the same for the second model: why limit the wild bird species to Passeriformes and Columbiformes? What is the rationale behind this??
Response 5: Very interesting question! Literature and observational data confirm that members of the Passeriformes and Columbiformes orders function as key synanthropic reservoirs and transmitters of avian influenza virus in the context of internal (within-country) disease spread.
Reaction to response 5: but you are very selective and biased in this, because there is also substantial literature indicating that other wild bird species, among which are Anseriformes and Charadriformes, that spread the virus within a country, so this seems not a good choice.
Author Response
Comments 3: Line 99-106: … and domestic ducks…..Please explain and argument why you would limit this to only domestic ducks?? There are other poultry species (layers, broilers, turkeys…..) that have been infected by HPAI H7 subtype. I see now plausible reason why you would limit the study to only domestic ducks.
Response 3: Thank you for raising this question. Notably, in Russia and East Asian regions, domestic ducks (Anatidae family) are commonly reared in free-range systems. This husbandry method renders them a pivotal node in the epidemiological transmission chain, owing to their increased potential for interaction with wild and synanthropic bird populations.
Reaction to response 3: it is true that outdoor-raised poultry has an increased potential for interaction with wild bird populations. But chickens in East Asian regions are also raised in open sheds with potential contact with wild birds. Furthermore, in West-European countries, poultry farms with chickens and domestic ducks get also infected although there is no physical contact with wild birds because the poultry is housed in closed poultry units. So in my view there is no reasonable argument to limit the analysis to only domestic ducks.
Response to reaction to response 3: Thank you for your interest in the study. We understand your question, but would like to emphasise that this research focuses on analysing the spread of avian influenza by wild birds. Our outbreak analysis identified Anseriformes as the predominant order, with ducks (Anatidae) as the dominant species. Since this family includes both wild and domestic ducks, we did not separate these data — in Russia, domestic ducks (given their specific husbandry practices) significantly influence the likelihood of pathogen introduction. In contrast, the Galliformes order accounts for only 3 % of wild bird outbreaks (Figure 3A), which does not substantially affect the overall introduction pattern.
Comments 5: Line 122-126: and the same for the second model: why limit the wild bird species to Passeriformes and Columbiformes? What is the rationale behind this??
Response 5: Very interesting question! Literature and observational data confirm that members of the Passeriformes and Columbiformes orders function as key synanthropic reservoirs and transmitters of avian influenza virus in the context of internal (within-country) disease spread.
Reaction to response 5: but you are very selective and biased in this, because there is also substantial literature indicating that other wild bird species, among which are Anseriformes and Charadriformes, that spread the virus within a country, so this seems not a good choice.
Response to reaction to response 5: We are grateful for your question and apologise for any confusion arising from our initial response. In the framework of the second model, we indeed considered wild avian species, including the orders Anseriformes and Charadriformes. This information is addressed in sections 2.1 “The second model incorporated factors related to the presence of humans, domestic, migratory, and synanthropic birds...”, 2.4 “The "spread model" was constructed using only population factors, an approach that implicates migratory and synanthropic birds in the potential spread of the influenza virus ...”, and further substantiated by the data presented in Figure 8.
Round 3
Reviewer 2 Report
Comments and Suggestions for Authors
I have nothing to add